# Comparison of citation rates between Covid-19 and non-Covid-19 articles across 24 major scientific journals

**Michael D. Brandt**[1]*, **Sherief A. Ghozy**[2], **David F. Kallmes**[2], **Robert J. McDonald**[2☯], **Ramanathan D. Kadirvel**[2☯]

**1** Alix School of Medicine, Mayo Clinic Hospital, Rochester, Minnesota, United States of America,
**2** Department of Radiology, Mayo Clinic Hospital, Rochester, Minnesota, United States of America

☯ These authors contributed equally to this work.
* brandt.michael@mayo.edu

**Data Availability Statement:** All relevant data are within the paper and its Supporting Information files.

## Abstract

Covid-19 has been front and center in the global landscape since the beginning of 2020. In response, the scientific field has dedicated enormous amounts of resources to researching the virus and its effects. The number of times Covid-19 publications are being cited throughout the literature appears remarkably high but has not been directly compared to non-Covid-19 papers in the same journals over an extended period. In our study, we use Clarivate's Web of Science—Science Citation Index Expanded™ database to identify Covid-19 papers published in 24 major scientific journals over a period of 24 months from January 1, 2020 to December 31, 2021. We conduct our search using keywords "Covid-19", "coronavirus", and "sars-cov-2" to locate publications with these words in the title. We then quantify the number of citations these papers have received and compare rates to non-Covid-19 papers in the same journals over the same timeframe. We find that, across 24 open-access and subscription-based scientific journals, Covid-19 papers published in the past 2 years currently have a median citation rate of 120.79 compared to 21.63 for non-Covid-19 papers. When negative binomial regression is used to minimize the influence of other variables such as article number variation and field of research, Covid-19 papers have still experienced more than 80% increase in citations relative to non-Covid-19 papers. These novel findings demonstrate that Covid-19 papers are being cited at remarkably higher rates than non-Covid-19 articles contained within the same journals. This suggests that journal impact factor, which is a product of the number of citations that recently published articles receive, will likely be drastically influenced by the number of Covid-19 papers that a journal has included within its pages in the previous years.

## Introduction

In the early days of January 2020, the World Health Organization (WHO) published a bulletin that brought attention to 44 new cases of pneumonia of unknown etiology that had appeared at the end of December 2019 in Wuhan City, Hubei Province of China [1]. Regional cases

**Funding:** The authors received no specific funding for this work.

**Competing interests:** The authors have declared that no competing interests exist.

increased rapidly over the next weeks, and scientists worldwide began to take notice. By the end of January, more than 50 research papers had been published about the outbreak [2]. In the coming months, while researchers scrambled to learn more about the coronavirus that was now spreading internationally, the WHO again released a bulletin on March 11, 2020, officially classifying Covid-19 as a global pandemic [3]. For months after, cases and fatalities rose at staggering clips as governing bodies worldwide grappled with the best measures to contain the virus. Eventually, strict regulations and vaccine rollouts were implemented with enough success to begin to slow case growth [4]. Amidst all of this, there has been an explosion of peer-reviewed literature about Covid-19 as researchers work to uncover details such as structure, infectivity, spread, effects, prevention, and treatment of this novel virus. According to the Web of Science Science Citation Index Expanded™ (WOS), which includes more than 8,300 journals across 150 scientific disciplines, nearly 200,000 Covid-19 related papers were published or presented in 24 months between January 1, 2020 and December 31, 2021 [5].

Unsurprisingly, several seminal Covid-19 publications have been cited at incredibly high rates as researchers have turned to these papers to help guide their next steps [6–8]. However, this citation trend in Covid-19 literature has not been limited to only a select few articles. As Covid-19 remains in the spotlight, the existing Covid-19 body of literature continues to be heavily leaned upon by scientists hoping to expand the existing knowledge further.

The extent to which Covid-19 papers are being cited in selected top journals relative to non-Covid-19 papers in the same journals has yet to be explored in detail. To address this gap in the literature we use WOS to examine citation rates for Covid-19-related articles published between January 1, 2020 and December 31, 2021 across 24 major scientific journals. We compare these rates with citation rates for non-Covid-19 articles published in the same 24 journals over an identical time frame. We anticipate that these findings could be of value to journals editors and researchers when considering efforts for future publications. In addition, we expect these findings to shed light on the influence that the current influx of Covid-19 literature will have on Journal Impact Factor (JIF) in the years ahead.

## Materials and methods

### Journal selection

Using information provided in Clarivate's 2020 Journal Citation Reports® [9] published on June 30, 2021, the top three journals by impact factor were selected from eight scientific categories as defined by WOS. Disciplines were selected based on the likelihood of having high relevancy to Covid-19 and, therefore, a sufficient volume of Covid-19-related papers in that field's premier journals. A baseline of 15 Covid-19 articles was set as a requirement for a journal to be included. If a top three journal did not meet this baseline, the next highest journal by impact factor from that field was included. The disciplines chosen were respiratory, cardiology, immunology, radiology, microbiology, gastroenterology and hepatology, general and internal medicine, and multidisciplinary sciences. With three journals each from eight fields, a total of 24 journals were selected (Table 1).

### Data collection and sample randomization

The WOS search criteria were customized to include documents categorized as "Article" published between January 1, 2020 and December 31, 2021. The database tags papers as articles if they meet the following criteria: "Reports of research on original works. Includes research papers, features, brief communications, case reports, technical notes, chronology, and full papers that were published in a journal and/or presented at a symposium or conference" [5]. In addition, we included papers categorized by Web of Science as "Review" to capture meta-

**Table 1. Summary of 24 selected journals.**

| Journal | Specialty | COVID-19 article | | Impact factor | Publisher | Publication mode (open-access, subscription-based, or hybrid) | Country | Number of issues/year |
|---|---|---|---|---|---|---|---|---|
| | | Yes | No | | | | | |
| | | Articles number | Articles number | | | | | |
| Nature | Multidisciplinary Sciences | 145 | 1929 | 49.962 | Nature Research | Hybrid | Germany | 51 |
| Nature Communications | Multidisciplinary Sciences | 428 | 12449 | 14.919 | Nature Research | Open-access | Germany | Continuous publishing |
| Science | Multidisciplinary Sciences | 145 | 1479 | 47.728 | Amer Assoc Advancement Science | Hybrid | USA | 51 |
| JAMA | Medicine, General, and Internal | 54 | 350 | 56.274 | Amer Medical Assoc | Subscription-based; free to public after 6 months | USA | 48 |
| Lancet | Medicine, General, and Internal | 89 | 382 | 79.323 | Elsevier Science Inc | Hybrid | USA | 52 |
| NEJM | Medicine, General, and Internal | 98 | 579 | 91.253 | Massachusetts Medical Soc | Hybrid | USA | 52 |
| Cell Host and Microbe | Microbiology | 53 | 217 | 21.023 | Cell Press | Subscription-based; free to public after 12 months | USA | 12 |
| Clinical Infectious Diseases | Microbiology | 569 | 1837 | 9.079 | Oxford Univ Press Inc | Hybrid | USA | 24 |
| Nature Microbiology | Microbiology | 27 | 256 | 17.745 | Nature Research | Hybrid | Germany | 12 |
| European Journal of Nuclear Medicine and Molecular Imaging | Radiology | 43 | 685 | 9.236 | Springer | Hybrid | USA | 12 |
| Radiology | Radiology | 75 | 497 | 11.105 | Radiological Soc North America | Hybrid | USA | 12 |
| Ultrasound in Obstetrics Gynecology | Radiology | 27 | 355 | 7.299 | Wiley | Hybrid | USA | 12 |
| Journal of Am Col Cardiology | Cardiology | 39 | 673 | 24.093 | Elsevier Science Inc | Hybrid | USA | 50 |
| Circulation | Cardiology | 22 | 593 | 29.690 | Lippincott Williams and Wilkins | Hybrid | USA | 50 |
| European Heart Journal | Cardiology | 20 | 524 | 29.983 | Oxford Univ Press | Hybrid | England | 24 |
| Gastroenterology | Gastroenterology and Hepatology | 27 | 523 | 22.682 | WB Saunders Co-Elsevier Inc | Hybrid | USA | 12 |
| Gut | Gastroenterology and Hepatology | 35 | 498 | 23.059 | BMJ Publishing Group | Hybrid | England | 12 |
| Journal of Hepatology | Gastroenterology and Hepatology | 17 | 452 | 25.083 | Elsevier | Hybrid | Netherlands | 12 |
| Am J of Respiratory and Crit Care Med | Respiratory | 30 | 371 | 21.405 | Amer Thoracic Soc | Hybrid | USA | 24 |
| European Respiratory Journal | Respiratory | 40 | 457 | 16.671 | European Respiratory Soc Journals Ltd | Hybrid | England | 12 |
| Lancet Respiratory Medicine | Respiratory | 67 | 118 | 30.700 | Elsevier Sci Ltd | Hybrid | England | 12 |
| Immunity | Immunology | 48 | 279 | 31.745 | Cell Press | Hybrid | USA | 12 |
| Nature Immunology | Immunology | 26 | 249 | 25.606 | Nature Research | Hybrid | Germany | 12 |
| Nature Reviews Immunology | Immunology | 23 | 111 | 53.106 | Nature Research | Subscription-based | Germany | 12 |

analyses and systematic reviews as well. All articles published within the 24-month time frame were selected to produce an average citation rate for the journal itself over that period. Next, articles without the terms "Covid-19", "coronavirus", or "SARS-CoV-2" in the title were selected to provide an average citation rate for only non-Covid-19 articles in the journal. Finally, articles containing the keywords "Covid-19", "coronavirus", or "SARS-CoV-2" in the title were selected to provide an average citation rate for Covid-19-related articles. In this way, three separate citation averages were gathered for each journal: 1) all articles 2) non-Covid-19 articles 3) only Covid-19 articles.

In many journals, there was discrepancy between the number of Covid-19 and non-Covid-19 articles. To establish a comparison that was more evenly matched in terms of article volume, randomization software on Microsoft Excel was used to select a sample of non-Covid-19 papers from each journal to create a 1:1 comparison of citation rates between non-Covid and Covid papers. A single average citation rate was obtained from this smaller sample as well.

## Data analysis

Data were analyzed with R software version 4.1.2 [10] using the packages (Rcmdr) [11] and (glm2) [12]. Significance was considered when the P-value was < 0.05. Median and range were used to represent continuous variables (not normally distributed) while we used frequencies and percentages to represent categorical variables. The skewness and Kurtosis tests were used for testing the normal distribution of continuous variables. We estimated the effect of Covid-19 subject on citation counts using a negative binomial regression model. The negative binomial regression model was selected over a linear regression model because it resulted in a better fit to the data and was more appropriate for count data. The negative binomial regression model is similar to the Poisson regression model (for count data) except that it performs better with data over-dispersion [13, 14]. The model was also used to assess any differences in citation rates attributed to the field category. Finally, the model was adjusted to account for the discrepancy in volume between non-Covid-19 papers and Covid-19 papers.

## Results

Comparisons between non-Covid-19 and Covid-19 articles including all fields are shown in Table 2. The median citation rate at two years for Covid-19 articles in the top journals across all eight fields is 120.79 (p = <0.001). For non-Covid-19 articles, the median citation rate is 21.63 (p = < 0.001). This equates to a Covid-19: non-Covid-19 ratio of 5.58 citations per article. When comparing Covid-19 papers and the 1:1 randomized sample of non-Covid-19 papers, the median citation decreases from 21.63 to 20.1 for non-Covid-19 citations and the citation ratio between Covid-19 and non-Covid-19 papers climbs to 6.01 (p = <0.001).

A negative binomial regression model was used to assess for potential confounding variables (Table 3). Only Covid-19 articles within the Medicine, General, and Internal category were significantly affected by the field itself with a 28% boost in citation rate relative to non-Covid-19 articles (p = 0.029, 95%CI [0.13–2.43]). No other fields accounted for significant difference in citation rates amongst Covid-19 and non-Covid-19 papers published within the same field. When controlling for influence of field categorization and article numbers, Covid-19 papers received 84% more citations than non-Covid-19 papers in the non-randomized sample (p<0.001, 95% CI [1.22–2.45]). When the randomized 1:1 sample was analyzed, Covid-19 papers still received 82% more citations than non-Covid-19 papers (p<0.001, 95% CI [1.20–2.43]).

**Table 2. Quantity of citations for Covid-19 and non-Covid-19 articles with median and range values across all fields.**

| Variables | | COVID-19 article | | | | Total | | P-value |
|---|---|---|---|---|---|---|---|---|
| | | No | | Yes | | (N = 28,010) | | |
| | | (n = 25,863) | | (n = 2,147) | | | | |
| | | Count | % | Count | % | Count | % | |
| Field | Cardiology | 1,790 | 6.9 | 81 | 3.8 | 1,871 | 6.7 | 0.006* |
| | Gastroenterology Hepatology | 1,473 | 5.7 | 79 | 3.7 | 1,552 | 5.5 | |
| | Immunology | 639 | 2.5 | 97 | 4.5 | 736 | 2.6 | |
| | Medicine, General, and Internal | 1,311 | 5.1 | 241 | 11.2 | 1,552 | 5.5 | |
| | Microbiology | 2,310 | 8.9 | 649 | 30.2 | 2,959 | 10.6 | |
| | Multidisciplinary Sciences | 15,857 | 61.3 | 718 | 33.4 | 16,575 | 59.2 | |
| | Radiology | 1,537 | 5.9 | 145 | 6.8 | 1,682 | 6.0 | |
| | Respiratory | 946 | 3.7 | 137 | 6.4 | 1,083 | 3.9 | |
| Variables | | Median | Range | Median | Range | Median | Range | P-value |
| Average citations | | 21.63 | 6.02–83.63 | 120.79 | 28.72–824.44 | 52.815 | 6.02–824.44 | < 0.001* |
| Average citations–random sample | | 20.1 | 6.99–79.31 | 120.79 | 28.72–824.44 | 53.935 | 6.99–824.44 | < 0.001* |

*Statistically significant

## Discussion

We used WOS to determine the difference in rates at which Covid-19 and non-Covid-19 articles from 24 top medical journals are being cited. Looking at all categories and journals that were included, Covid-19 articles published between January 1, 2020 and December 31, 2021 are being cited at considerably higher rates than the non-Covid-19 papers. Across eight selected fields, the median citation for a Covid-19 paper approaches six times that of a non-Covid-19 paper within the same journals. This holds even when article volume is equated using a 1:1 sample of non-Covid to Covid articles as median citation is six times greater for the Covid-19 articles. Using a negative regression model to analyze the entire data, Covid-19 papers have 84% more citations than non-Covid-19 papers when controlling for field and article number discrepancies. This number dips only slightly to 82% when the smaller,

**Table 3. Negative binomial regression output reporting independent variable effects on citation count¶.**

| Predictors | Non-randomized sample | | | | Randomized sample | | | |
|---|---|---|---|---|---|---|---|---|
| | IRR | 95% Confidence Interval | | P-value | IRR | 95% Confidence Interval | | P-value |
| | | Lower | Upper | | | Lower | Upper | |
| Cardiology | Reference | | | | | | | |
| Respiratory | -0.17 | -1.32 | 0.98 | 0.771 | -0.31 | -1.46 | 0.84 | 0.592 |
| Radiology | -0.76 | -1.92 | 0.40 | 0.197 | -0.82 | -1.97 | 0.34 | 0.167 |
| Multidisciplinary Sciences | 0.53 | -0.72 | 1.77 | 0.407 | 0.48 | -0.76 | 1.72 | 0.449 |
| Microbiology | -0.22 | -1.38 | 0.93 | 0.702 | -0.29 | -1.45 | 0.86 | 0.619 |
| Medicine, General, and Internal | 1.28 | 0.13 | 2.43 | 0.029* | 1.20 | 0.05 | 2.34 | 0.041 |
| Immunology | 0.11 | -1.03 | 1.26 | 0.846 | 0.15 | -1.00 | 1.30 | 0.800 |
| Gastroenterology Hepatology | -0.26 | -1.41 | 0.89 | 0.659 | -0.33 | -1.48 | 0.82 | 0.572 |
| Non-COVID-19 article | Reference | | | | | | | |
| COVID-19 article | 1.84 | 1.22 | 2.45 | <0.001* | 1.82 | 1.20 | 2.43 | <0.001* |

¶Adjusted for articles' number

*Statistically significant

randomized sample is compared. 'Amongst all fields, only articles categorized as Medicine, General, or Internal according to WOS see a bump (28%) in citation rates that can be attributed to the field itself. Thus, an article's focus on Covid-19 seems to be a primary driver of increased citations for these articles compared to those that do not deal with a Covid-19 related topic.

Reasons behind the major increase in citations for Covid-19 articles seem straightforward. Covid-19 has dominated global focus since the onset of 2020, affecting over 200 countries across the world [15]. As such, researchers are eager to add their contributions to what is known and what can be done to combat Covid-19 and, to do this, are citing earlier works to support their approaches. This earnest for more knowledge is further bolstered by increased government funding for Covid-19 research. In the United States, the Coronavirus Aid, Relief, and Economic Security (CARES) Act was signed into law on March 27, 2020 and allocates 940 million dollars to the National Institute of Health (NIH) to be used for funding Covid-19 research [16]. Thus, more funding opportunities are available at the researcher level for research that focuses on Covid-19. Another potential reason for the spike in Covid-19 citations is the exhibited capacity of the virus to mutate rapidly. As new variants appear, researchers are offered the chance to publish on new epidemiological or variant characterization studies.

Previous studies have looked at citation rates for preprints of Covid-19 papers [17] and quality of evidence contained within published Covid-19 articles [18, 19]. Additionally, citation rate for Covid-19 papers has been assessed previously without direct comparison to non-Covid-19 papers in the same journals [20]. To our knowledge, this study is the first to quantify the rate at which a large volume of peer-reviewed Covid-19 articles are being cited, on average, in major medical journals and compare these results to non-Covid-19 articles in the same journals. In doing this, we demonstrate the sharp contrast between citation rates for Covid-19 and non-Covid-19 articles published in 24 months during the rise and height of the global pandemic. While our research reveals the substantial degree to which Covid-19 articles are being cited in top journals relative to non-Covid-19 articles, the effect on journals themselves remains to be seen.

One way that these effects may be observed is through the influence that Covid-19 papers will have on JIF. JIF is a commonly used surrogate to determine journal excellence and is calculated by dividing number of citations in the current year for articles published in the previous two years by total number of articles published in that journal during the previous two years [21]. Therefore, it is a direct measure of how many citations recently published articles in a given journal receive. As we have shown, Covid-19 papers across all selected fields are being cited at a vastly increased rate compared to non-Covid-19 papers within the same journals. We would estimate that as Covid-19 papers continue to flood the literature across all scientific fields, so long as Covid-19 continues to hold the global spotlight, Covid-19 papers will continue to be cited, on average, at much higher levels than non-Covid-19 papers. We anticipate that JIFs will be affected in coming years which could change the landscape for how journal excellence is determined in the future.

## Limitations

For the scope of this paper, we used papers categorized as "Article" or "Review" in our WOS search to ensure that we were focusing on original works and evidence-based literature. This means that other publications such as letters, editorials, perspectives, and opinions were not included in our search criteria which leaves a large number of publications out of the study. Inclusion of these would undoubtedly have increased the volume of papers and affected the results, though the direction or magnitude of change is unclear. Another limitation of this

study is that open-access versus subscription-based journals were not filtered separately during the WOS search. Thus, data include a combination of open-access and subscription-based publications in the non-Covid-19 articles. This would seemingly be an impactful factor in how often these articles are being cited. However, a recent meta-analysis showed that the advantage of open-access is debatable with many studies showing no difference in citation rates andquality and heterogeneity concerns posing challenges for generalization [22].

Finally, though WOS is a comprehensive and highly regarded database, it is not without shortcomings. Most notably, its categorization of a publication as "Article" seems imperfect. To examine this more closely, we randomly selected three journals included for this paper and manually searched each journal's website for Covid-19-related articles that the journal itself had categorized as an original piece of research over three months. We compared this to the papers under "Article" that WOS captured over the same 3-month period for that journal. We found that the WOS database included 76% of the Covid-19 papers contained within the journals themselves. Further, 91% of the papers under the "Article" category in the database were classified as original works by the journals themselves. In our sample, the database captured about three-quarters of the papers that should be included but is more precise in labeling only reports of research on original work as "Article". Though our method of locating Covid-19 and non-Covid-19 papers in the selected journals was more expedient, manual scouring of the journals themselves over the 24 months would have optimized both the sensitivity and specificity of locating and comparing the most comprehensive list of publications possible.

## Supporting information

**S1 Data.**
(XLSX)

## Author Contributions

**Conceptualization:** Michael D. Brandt, David F. Kallmes, Robert J. McDonald, Ramanathan D. Kadirvel.

**Data curation:** Michael D. Brandt.

**Formal analysis:** Sherief A. Ghozy.

**Investigation:** Michael D. Brandt.

**Methodology:** Michael D. Brandt, David F. Kallmes, Robert J. McDonald, Ramanathan D. Kadirvel.

**Software:** Sherief A. Ghozy.

**Supervision:** David F. Kallmes, Robert J. McDonald, Ramanathan D. Kadirvel.

**Writing – original draft:** Michael D. Brandt.

**Writing – review & editing:** Michael D. Brandt, Sherief A. Ghozy, David F. Kallmes, Ramanathan D. Kadirvel.

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
