## [Decision Letter · Decision Letter 0]

2 May 2022

PONE-D-22-05834Comparison of citation rates between Covid-19 and non-Covid-19 articles across 18 major medical journalsPLOS ONE

Dear Dr. Brandt,

Thank you for submitting your manuscript to PLOS ONE. After careful consideration, we feel that it has merit but does not fully meet PLOS ONE’s publication criteria as it currently stands. Therefore, we invite you to submit a revised version of the manuscript that addresses the points raised during the review process.

Please submit your revised manuscript by Jun 09 2022 11:59PM. If you will need more time than this to complete your revisions, please reply to this message or contact the journal office at plosone@plos.org. Please include the following items when submitting your revised manuscript:A rebuttal letter that responds to each point raised by the academic editor and reviewer(s). You should upload this letter as a separate file labeled 'Response to Reviewers'.A marked-up copy of your manuscript that highlights changes made to the original version. You should upload this as a separate file labeled 'Revised Manuscript with Track Changes'.An unmarked version of your revised paper without tracked changes. You should upload this as a separate file labeled 'Manuscript'.

We look forward to receiving your revised manuscript.

Kind regards,

Ayman Elbehiry

Academic Editor

PLOS ONE

Journal Requirements:

(The funders had no role in study design, data collection and analysis, decision to publish, or preparation of the manuscript.) 

4. Please provide additional details regarding participant consent. In the ethics statement in the Methods and online submission information, please ensure that you have specified what type you obtained (for instance, written or verbal, and if verbal, how it was documented and witnessed). If your study included minors, state whether you obtained consent from parents or guardians. If the need for consent was waived by the ethics committee, please include this information.

Reviewers' comments:

Reviewer's Responses to Questions

**Comments to the Author**

1. Is the manuscript technically sound, and do the data support the conclusions?

Reviewer #1: Yes

Reviewer #2: No

2. Has the statistical analysis been performed appropriately and rigorously? 

Reviewer #1: Yes

Reviewer #2: No

3. Have the authors made all data underlying the findings in their manuscript fully available?

Reviewer #1: No

Reviewer #2: No

4. Is the manuscript presented in an intelligible fashion and written in standard English?

Reviewer #1: Yes

Reviewer #2: No

5. Review Comments to the Author

Reviewer #1: - The scientific value of the research is low, but the authors followed a good research methodology, data analysis , as well as the unique research idea that enriched the article.

- Judging in this research depends on the availability of data about scientific journals that have been cited, which supports the results, and therefore the authors must attach the list of scientific journals that were included in the study, (it will be useful if added to the methodology), even if that would benefit the journals but it is not considered advertisement for them, it is their right, and in the same time it is essential document for the publishing this article

Reviewer #2: PLOS ONE

Comparison of citation rates between Covid-19 and non-Covid-19 articles across 18 major medical journals

PONE-D-22-05834

Thank you for asking me to review the above-titled manuscript. The topic is interesting. However, there are significant problems in the manuscript.

Abstract: 1) Line 29- What are these major journals? 2) Line 30 - start starting and end dates; why only 18 months and not up to 31 December 2021? 3) How did you search the WOS? 4) Were these non-covid and covid papers published in the same journal? 5) Were these open-access journals or subscription-based journals? Did you balance between them? 6) What were the JIFs of these journals or the median IQR? 7) Lines 34-35-- look strange because all papers on COVID-19 in subscription-based journals were available free of charge. I cannot see any explanation in your discussion 8) JIF is measured every 2 years, not 18 months. 9) Lines 37-39- on what basis did you come. 10) What was the percentage of Covid versus non-covid in each journal selected.

Introduction - 1) China declared a novel virus causing this disease by December 2019. 2) Lines 56, page 3: Why not up to 31 December 2021? to include vaccination research as well? 3) Give references/citations (lines 57-63). 4) Not clear what is the problem that triggered the study? 5) What is the purpose of the study? We all know that citations are most likely higher for COVID-19 papers, you need to dig deeper into this. 6) What is your research question?

Methods- 1) Line 75- was the number of articles on Covid-19 and non-Covid balanced? How? 2) Line 72-Why only top JIF journals? 3) Why did you not target the top JIF (Q1), the middle JIF journals (Q2-Q3) and those with low JIF (Q4)? 4). Why were Journals on Neurology and those on Gastroenterology and Hepatology not included? 5) What do you mean by Multidisciplinary Science journals? Line 78? 6) How many journals were included for each field? STATE THE JOURNALS INCLUDED IN EACH FIELD. 7) WHAT ARE THESE THREE TOP JOURNALS? Were these on Microbiology and or Virology?

Methods- It is unusual and strange that the authors do not provide any details about their work. Research must be transparent.

Methods- Lines 80-85 We know that this is a limitation in the classification of articles in WoS. Did you identify them in the analysis and show numbers of research, numbers of case report, numbers of brief communication etc, and you should define each category.

Methods- Lines 87 (page 4) and linnes 88-90 (page 5) NOT CLEAR.

Methods 94-96 - The aim should be EQUAL NUMBERS of Covid-19 and Non-Covid papers not 2:1. The number of Non-Covid is usually small, so accept it as the basic number and randomise for an equivalent number of COVID-19 papers. This should be the way.

METHODS- WHY THE AUTHORS DID NOT INCLUDE FUNDING AND COMPARE BETWEEN THE COVID-19 PAPERS and NON-COVID PAPERS?

Methods- What were the countries and institutes that produced these papers in both groups?

Methods: WHY WERE THE ALTMETRIC SCORES NOT COMPARED? As well?

English needs editing. Several statements are not clearly stated. Also, statements like "This was done..."

Table 1- 1) What is the number of journals in each category "Multidisciplinary, Cardiology etc" 2) What is the NAME of the Journal in each category? 3) What is the JIF of each journal? 4) State the number of papers from each journal in Covid-19 and Non-Covid 19.

Table- You need a new table summarising key topics raised in each category and numbers in both groups.

Discussion- 1) Poorly written. 2) Why do lines 141-155 have no citations? 2) Funding should be compared between the two groups in the study; SEE COMMENTS UNDER METHODS? 3) We need to know with journals were open access and which one was subscription-based paper.

Discussion- No explanation was given to the 10 times issue of open access journals. We know that all papers on COVID-19 were available free of charge and this is an important cause for the high citations of the COVID-19 papers (not discussed). BUT it does not explain the authors' claim for 10 times issue (also missing).

Limitations: 1) Articles are not only research and what is stated is not accurate. 2) Are case reports evidence-based? 3) Talking about evidence-based work- The top in this are Systematic Reviews and meta-analyses. MMy question, Why these two groups were excluded? Lines 191.

What are these three journals- state their Names. For all journals, you must state title, company/publisher, JIF, open-access or subscription-based, country, and the number of issues per year, the year it was issued. Show the number of papers for each group This could be in a table.

Conclusion- should be rewritten.

References- Should be improved.

6. PLOS authors have the option to publish the peer review history of their article (what does this mean?). If published, this will include your full peer review and any attached files.

Reviewer #1: **Yes: **Dr Mustafa Mohammed Mustafa

Reviewer #2: No

---

## [Author Response · Author response to Decision Letter 0]

13 Jun 2022

Comments also included in separate attached file title "Response to Reviewers"

Response to Reviewer Comments

Reviewer #1: The scientific value of the research is low, but the authors followed a good research methodology, data analysis, as well as the unique research idea that enriched the article.

We appreciate the positive review of our study

1. Judging in this research depends on the availability of data about scientific journals that have been cited, which supports the results, and therefore the authors must attach the list of scientific journals that were included in the study, (it will be useful if added to the methodology), even if that would benefit the journals but it is not considered advertisement for them, it is their right, and in the same time it is essential document for the publishing of this article. Thank you for bringing this to our attention. We created an additional table to list all journals included in the study. We will also include all data gathered from each journal in our data set which will be uploaded as a Supporting Information File.

Reviewer #2: Thank you for asking me to review the above-titled manuscript. The topic is interesting. However, there are significant problems in your manuscript.

We thank you for your interest in the study and willingness to provide helpful feedback.

1. Line 29- What are these major medical journals? Because there are 24 journals, we believe that it would detract from the focus of the abstract to list them individually. They are listed separately in Table 1 in the methodology section.

2. Line 30- Start and end dates; Why only 18 months and not up to 31 December 2021? Start and end dates have now been included. Data has been expanded to include 24 months up to 31 December 2021. 

3. How did you search the WOS? The search was conducted by searching for all titles that had “Covid, “sars-cov-2”, or “coronavirus” in a given journal and obtaining a single average citation for these papers. We then repeated the process for all titles that did not have any of the aforementioned keywords and obtained a single citation for these papers as well. The process was repeated individually for each journal. We have included an additional sentence explaining briefly how the search was conducted.

4. Were these non-Covid and Covid papers published in the same journal? We have adjusted the wording to make it clearer that these papers were published within the same journal.

5. Were these open-access journals or subscription-based journals? Did you balance between them? We have added a line stating that both open-access and subscription-based journals were included. The great majority of the included journals employ a hybrid model with options for both open-access and subscription-based publishing. WOS does allow for filtering of open-access articles but fails to filter for articles that were originally published as subscription-based but become free to the public after a specified period of time has elapse, as is the protocol for several of the journals that we searched. Further, a recent meta-analysis showed the advantage of open-access is debatable with many studies showing no difference, with quality and heterogeneity concerns posing challenges for generalization. 

Reference: 

Langham-Putrow A, Bakker C, Riegelman A (2021). Is the open access citation advantage real? A systematic review of the citation of open access and subscription-based articles. PLOS ONE 1e0253129. https://doi.org/10.1371/journal.pone.0253129

6. What are the JIFs of these journals or the median IQR? JIFs for all journals now included as part of additional table (Table 1).

7. Lines 34-35—look strange because all COVID-19 in subscription-based journals were available free of charge. I cannot see any explanation in your discussion. Wording has been adjusted to clarify this point.

8. JIF is measured every 2 years, not 18 months. Data now reflects 2-year period from 1/1/2020 to 31/12/2021 rather than 18 months as previously.

9. Lines 37-39- on what basis did you come? We are arriving at this suggestion based on our findings that Covid-19 papers are being cited at much higher rates than non-Covid papers within the same journals. Therefore, we expect that journals will see their JIF inflate in the coming years due given that they have been publishing (Covid) articles that are, on average, more highly cited than articles that are unrelated to Covid. We have adjusted the wording to make this point more salient.

10. What was the percentage of Covid versus non-Covid in each journal selected? Total number of papers of each type now included in Table 1. 

11. China declared a novel virus causing this disease by December 2019. Wording adjusted to make this point clearer.

12. Lines 56, page 3: Why not up to 31 December 2021 to include vaccination research as well? Data now includes all publications up to this date.

13. Give references/citations (lines 57-63). Citation for these lines has now been added.

14. Not clear what is the problem that triggered the study? What is the purpose of the study? We all know that citations are most likely higher for COVID-19 papers, you need to dig deeper into this. What is your research question? We agree that it is widely known that Covid-19 papers are cited at a higher rate across the literature. However, we note that there is a gap in the literature comparing the rate at which Covid-19 papers in top journals are being cited to the rate at which non-Covid articles in the same journals are being cited. By doing this direct comparison we intend to answer the question “How much different is the rate of citation for Covid articles versus non-Covid articles published within the same journals across 24 of the top journals spanning the scientific field?” Though not the main focus of our paper, we believe that these data will be helpful in the future when determining the utility of JIF to quantify journal excellence and prestige in a post-Covid landscape. 

15. Methods-1) Line 75- was the number of articles on Covid-19 and non-Covid balanced? How? Yes, in Tables 2 and 3 we include both absolute numbers and a balanced 1:1 random sample of Covid and non-Covid papers.

16. Line 72- why only top JIF journals? Why did you not target the top JIF (Q1), the middle JIF journals (Q2-Q3) and those with low JIF (Q4)? We targeted only top JIF journals from 8 fields with the belief that these 24 journals and the >20000 articles within would provide adequate sample size to highlight the different rates at which Covid-19 and non-Covid-19 articles from the same journals are being cited. 

17. Why were journals on Neurology and those on Gastroenterology and Hepatology not included?

Top journals by JIF in Neurology did not meet inclusion criteria of minimum number of Covid articles per WOS search. Thank you for the suggestion of adding Gastroenterology and Hepatology. Data from the top 3 journals in this category have been added. 

18. What do you mean by Multidisciplinary Science journals? This is the categorization that is assigned by WOS to journals that publish broadly across the field of science.

19. How many journals were included for each field? State the journals included in each field. What are these three top journals? Were these on Microbiology and or Virology? It is unusual and strange that the authors do not provide any details about their work. Research must be transparent. Table 1 has been added which lists all journals (3 from each field) and will be included in the methodology section. These journals were not specific to Microbiology or Virology. We have added Microbiology as a separate category and included 3 journals from this category as well.

20. Methods- lines 80-85- We know that this is a limitation in the classification of articles in WOS. Did you identify them in the analysis and show numbers of research, numbers of case report, numbers of brief communication etc, and you should define each category. Unfortunately, WOS classifies each of these article types under a single “Article” categorization. Distinguishing between paper types would require manual individual extraction of each paper which is not feasible given volume of included papers. Further, as our main objective is to compare citation rates between Covid and non-Covid articles within the same journals, we believe that focus on sub-categorization does not advance the paper toward the main objective.

21. Lines 87 (page 4) and lines 88-90 (page 5) not clear. Wording has been adjusted for clarity

22. Methods 94-96- The aim should be equal numbers of Covid-19 and Non-Covid papers not 2:1. The number of Non-Covid is usually small, so accept it as the basic number and randomize for an equivalent number of Covid-19 papers. Thank you for this recommendation. We have updated the date to compare a 1:1 sample rather than 2:1. 

23. Why did the authors not include funding and compare between the Covid-19 papers and non-Covid papers? Determining funding would require manual extraction as WOS does not allow for filtering by funding. This is not feasible given number of papers.

24. What were the countries and institutes that produced these papers in both groups? Similarly, this would require manual extraction from each paper which was not feasible given volume.

25. Methods: Why were the altimetric scores not compared as well? Given our specific objective of comparing citation rates in Covid and non-Covid articles within top selected journals based on JIF, we believe that this would not influence the objective.

26. English needs editing. Several statements are not clearly stated. Also, statements like “This was done…” We reworded any statements that appeared unclear and removed “This was done..” where present

27. Table 1 1) What is the number of journals in each category “Multidisciplinary, Cardiology etc”. What is the name of the journal in each category? What is the JIF of each journal? State the number of papers from each journal in Covid-19 and Non-Covid-19. We have created a new table (now Table 1) which provides each of these details. 

28. Table- you need a new table summarizing key topics raised in each category and numbers in both groups. Given the volume of papers, it would not be feasible to go through each paper to determine key topics as there is not a set way to do this with the WOS search function.

29. Discussion- Poorly written. Discussion reworked to improve fluidity and clarity

30. Why do lines 141-155 have no citations? These lines are in reference to the data accrued from this study. We have added references to the tables for clarity.

31. We need to know which journals were open access and which one was subscription-based paper. No explanation was given to the 10 times issue of open access journals. We know that all papers on Covid-19 were available free of charge and this is an important cause for the high citations of the Covid-19 papers (not discussed). But it does not explain the authors’ claim for 10 times issue (also missing). Table 1 now includes the mode of publication for each journal. In the updated data (Table 3), we have added “Field” as a categorical variable which should help address the 10 times issue and we also added an additional 6 months’ worth of publications which should both help to make the data more logical. We have added the open-access versus subscription-based issue to our limitations as this could impact citations but feel that the recently published large meta-analysis (reference included in response to comment number 5) provides support that there is not clear-cut evidence to suggest that open-access publications will inherently be cited at a greater rate than subscription-based publications. 

32. Limitations- Articles are not only research and what is stated is not accurate. Are case reports evidence-based? Talking about evidence-based work- The top in this are Systematic Reviews and meta-analyses. Why were these two groups excluded? Line 191. WOS defines “Article” as “Reports of research on original works. Includes research papers, features, brief communications, case reports, technical notes, chronology, and full papers that were published in a journal and/or presented at a symposium or conference.” Your point that Systematic Reviews and Meta-Analyses are evidence-based work is well-made and these publications fall under the “Review” category for WOS. For the updated data, we included all papers under the “Review” category as well as the “Article” category. We have included a link to the WOS category distinction for reference.

- https://images.webofknowledge.com/images/help/WOS/hs_document_type.html

33. What are these three journals- state their names. For all journals, you must state title, company/publisher, JIF, open-access or subscription-based, country, and the number of issues per year. Show the number of papers for each group. This could be a table. Thank you for this helpful suggestion. Table 1 has been created to consolidate these pieces of information.

34. Conclusion- Should be rewritten. Conclusion was reworked for clarity and fluency

35. References should be improved. References updated for improved clarity and formatting

---

## [Editor Report · Decision Letter 1]

23 Jun 2022

Comparison of citation rates between Covid-19 and non-Covid-19 articles across 24 major scientific journals

PONE-D-22-05834R1

Dear Dr. Michael Brandt,

We’re pleased to inform you that your manuscript has been judged scientifically suitable for publication and will be formally accepted for publication once it meets all outstanding technical requirements.

Kind regards,

Ayman Elbehiry

Academic Editor

PLOS ONE
---

## [Editor Report · Acceptance letter]

18 Jul 2022

PONE-D-22-05834R1 

Comparison of citation rates between Covid-19 and non-Covid-19 articles across 24 major scientific journals 

Dear Dr. Brandt:

I'm pleased to inform you that your manuscript has been deemed suitable for publication in PLOS ONE. Congratulations! Your manuscript is now with our production department. 

Kind regards, 

on behalf of

Professor Ayman Elbehiry 

Academic Editor

PLOS ONE